# Adverse Effects of Angiotensin-Converting Enzyme Inhibitors in Humans: A Systematic Review and Meta-Analysis of 378 Randomized Controlled Trials

**DOI:** 10.3390/ijerph19148373

**Published:** 2022-07-08

**Authors:** Mingkwan Na Takuathung, Wannachai Sakuludomkan, Rapheephorn Khatsri, Nahathai Dukaew, Napatsorn Kraivisitkul, Balqis Ahmadmusa, Chollada Mahakkanukrauh, Kachathip Wangthaweesap, Jirakit Onin, Salin Srichai, Nida Buawangpong, Nut Koonrungsesomboon

**Affiliations:** 1Department of Pharmacology, Faculty of Medicine, Chiang Mai University, Chiang Mai 50200, Thailand; mingkwan.n@cmu.ac.th (M.N.T.); wannachai.yin@gmail.com (W.S.); veaw059@hotmail.com (R.K.); nahathai.miw@gmail.com (N.D.); 2Clinical Research Center for Food and Herbal Product Trials and Development (CR-FAH), Faculty of Medicine, Chiang Mai University, Chiang Mai 50200, Thailand; 3Faculty of Medicine, Chiang Mai University, Chiang Mai 50200, Thailand; napatsorn.krai@cmu.ac.th (N.K.); balqiz_a@cmu.ac.th (B.A.); chollada_m@cmu.ac.th (C.M.); kachathip_w@cmu.ac.th (K.W.); jirakit_on@cmu.ac.th (J.O.); salin_s@cmu.ac.th (S.S.); 4Department of Family Medicine, Faculty of Medicine, Chiang Mai University, Chiang Mai 50200, Thailand; nidalooknum@gmail.com; 5Musculoskeletal Science and Translational Research Center, Chiang Mai University, Chiang Mai 50200, Thailand

**Keywords:** angiotensin-converting enzyme inhibitors, ACE inhibitors, adverse effects, adverse drug reactions, safety information

## Abstract

Background: Although angiotensin-converting enzyme (ACE) inhibitors are among the most-prescribed medications in the world, the extent to which they increase the risk of adverse effects remains uncertain. This study aimed to systematically determine the adverse effects of ACE inhibitors versus placebo across a wide range of therapeutic settings. Methods: Systematic searches were conducted on PubMed, Web of Science, and Cochrane Library databases. Randomized controlled trials (RCTs) comparing an ACE inhibitor to a placebo were retrieved. The relative risk (RR) and its 95% confidence interval (95% CI) were utilized as a summary effect measure. A random-effects model was used to calculate pooled-effect estimates. Results: A total of 378 RCTs fulfilled the eligibility criteria, with 257 RCTs included in the meta-analysis. Compared with a placebo, ACE inhibitors were associated with an significantly increased risk of dry cough (RR = 2.66, 95% CI = 2.20 to 3.20, *p* < 0.001), hypotension (RR = 1.98, 95% CI = 1.66 to 2.35, *p* < 0.001), dizziness (RR = 1.46, 95% CI = 1.26 to 1.70, *p* < 0.001), and hyperkalemia (RR = 1.24, 95% CI = 1.01 to 1.52, *p* = 0.037). The risk difference was quantified to be 0.037, 0.030, 0.017, and 0.009, respectively. Conclusions: We quantified the relative risk of numerous adverse events associated with the use of ACE inhibitors in a variety of demographics. This information can help healthcare providers be fully informed about any potential adverse consequences and make appropriate suggestions for their patients requiring ACE inhibitor therapy.

## 1. Introduction

Angiotensin-converting enzyme (ACE) inhibitors are among the most widely prescribed drugs in the world, with hundreds of thousands of patients exposed to this drug class every year [1,2]. Ever since captopril became available in the 1980s [3], this drug class has been commonly used to treat cardiovascular disease, including hypertension, coronary artery disease, and heart failure, as well as other comorbid conditions, such as diabetic nephropathy [4,5,6,7]. ACE inhibitors act by inhibiting the activity of ACE, a key component of the renin–angiotensin–aldosterone system, thereby preventing the conversion of angiotensin I to angiotensin II [8,9]. Angiotensin II suppression leads to several physiologic consequences, such as relaxation of arteriolar vascular smooth muscle, decreased aldosterone secretion, and neurohormonal modulation, resulting in clinical benefits in a variety of cardiovascular diseases [10].

ACE inhibitors can cause adverse drug reactions, such as dry cough [11,12], and sometimes such reactions may result in the discontinuation of the drug [13,14]. Despite the remarkable growth in the use of ACE inhibitors worldwide, the degree to which ACE inhibitors increase the risk of adverse consequences remains uncertain. Most systematic reviews and meta-analyses have mainly focused on the efficacy of ACE inhibitors against certain conditions [15,16,17,18,19,20], whereas systematic analyses of adverse outcomes of ACE inhibitors are limited. Given the common use of ACE inhibitors worldwide, a systematic and quantitative assessment of the safety and tolerability of this drug class across a broad range of indications would be worthwhile, as it would give healthcare providers a comprehensive and precise awareness of all possible adverse drug reactions and enable them to appropriately advise their patients [21]. For this reason, we undertook a systematic review and meta-analysis of multiple randomized controlled trials (RCTs) of ACE inhibitors versus placebo, intending to provide greater insights into the safety of ACE inhibitors, irrespective of the specific diseases being studied.

## 2. Materials and Methods

In this study, we followed the Preferred Reporting Items for Systematic Reviews and Meta-Analyses (PRISMA) 2020 guidelines [22] and the PRISMA harms checklist [23]. The review protocol was developed in accordance with the PRISMA protocol (PRISMA-P) [24] and was prospectively registered at the PROSPERO international prospective register of systematic reviews in health and social care (CRD42021224281).

### 2.1. Search Strategy and Eligibility Criteria

An initial literature search was performed in three medical databases, namely PubMed, Web of Science, and Cochrane Library (last search: 15 November 2020), supplemented by a review of the reference lists of relevant articles for potentially eligible studies. Search strategies were conducted with the terms related to ACE inhibitors (i.e., angiotensin-converting enzyme inhibitors OR benazepril OR captopril OR cilazapril OR enalapril OR enalaprilat OR fosinopril OR lisinopril OR moexipril OR perindopril OR quinapril OR ramipril OR trandolapril) AND placebo, with no language restriction. All articles that were deemed relevant based on the title and abstract screening were retrieved and assessed with respect to eligibility criteria: (1) a study involving human subjects, (2) a study investigating the effect of an ACE inhibitor, (3) a study having a placebo-control group, and (4) a study having a report of adverse outcomes. Included articles were limited to an RCT (either a parallel or crossover) design. Observational studies, review articles, nonhuman studies (e.g., animal experiments), and other types of articles (e.g., case reports or expert opinion) were excluded. Studies were independently selected by at least two review authors, and the decision to include or exclude was arbitrarily based on consensus.

### 2.2. Data Extraction

We applied a hybrid approach consisting of prespecified adverse outcomes of interest (confirmatory approach) and additional common adverse outcomes being later identified (exploratory approach). Prespecified adverse outcomes of interest included four well-known adverse outcomes of ACE inhibitors, i.e., dry cough, hypotension, hyperkalemia, and angioedema. Additional adverse outcomes (either self-reported or actively sought) that were identified in more than three studies while conducting the systematic review were also included in the meta-analysis. The review authors did not group adverse outcome data in a composite measure. The data were not extracted as “0” unless reported as such in the original publication. A general statement indicating the absence of the event (for example, “no adverse events were identified”) was not extracted as a “0” event and such a statement was not included in further analysis. In the case that there were multiple reports from the same trial, the most complete and relevant dataset was used for analysis. For any unclear data, we made several attempts to e-mail the corresponding authors to ask for clarification.

### 2.3. Risk of Bias Assessment

Five domains of bias (i.e., bias arising from the randomization process, bias owing to deviations from intended interventions, bias caused by missing outcome data, bias in measurement of the outcome, and bias in selection of the reported result) were assessed following the most updated tool for assessing the risk of bias in RCTs (RoB2) [25]. Each RCT was judged to be at low or high risk of bias, or some concerns were raised based on the RoB2 algorithm. An R package and Shiny web app for visualizing risk of bias assessment (robvis) were applied to illustrate the results of the RoB2 assessment [26].

### 2.4. Statistical Analysis

For each adverse outcome, the extracted data were tabulated in a 2 × 2 contingency table. The relative risk (RR) and its corresponding 95% confidence interval (95% CI) were used as a summary effect measure. Pooled-effect estimates were calculated by means of a random-effects model, which reflected the systematic variation in estimates among studies. A funnel plot was applied to explore the potential of small-study effects, including publication bias, followed by the application of a linear regression test when there were 10 or more RCTs in a meta-analysis [27]. Statistical heterogeneity among studies was assessed using the Cochran’s *Q* test and the *I*^2^ statistic; *I*^2^ > 75% suggests high heterogeneity, whereas *I*^2^ < 40% indicates that statistical heterogeneity might not be important [28,29]. Substantial heterogeneity (defined as *I*^2^ > 50%) was explored using subgroup analyses and/or meta-regression, as appropriate, based on prespecified factors, i.e., drug name, drug dosage, and treatment duration. Sensitivity analyses were performed, with respect to the risk of bias, by excluding studies with a high risk of bias.

All statistical tests were performed in RStudio version 1.3.1093 (R Foundation, Vienna, Austria) for meta-analyses and data visualization, and a *p* value of < 0.05 was regarded to indicate statistical significance. A package ‘meta’ version 4.20-1 was executed to analyze meta-analyses. The package ‘ggplot2’ version 3.3.5 was used to construct heatmap visualization, and package ‘circlize’ version 0.4.13 was used to generate a chord diagram.

## 3. Results

Of the 9716 records that were retrieved from the three databases, a total of 378 RCTs fulfilled the eligibility criteria, with 257 RCTs included in the meta-analysis (Figure 1). Nineteen ACE inhibitors had been investigated in an RCT from 1979 to 2020 (Figure 2). The most widely studied ACE inhibitors included captopril (*n* = 41,263, from 65 studies), perindopril (*n* = 36,598, from 47 studies), ramipril (*n* = 18,615, from 39 studies), and enalapril (*n* = 10,357, from 107 studies) (Figure 2 and Figure 3). The efficacy and effectiveness of ACE inhibitors were investigated for several diseases, among which cardiovascular disease was the most widely studied condition (*n* = 97,872, from 246 studies), followed by cerebrovascular disease (*n* = 9,589, from 18 studies) and endocrine disorders (*n* = 7177, from 37 studies) (Figure 3). The median follow-up duration in the trials was 12 weeks (ranging from one-quarter of an hour to 10 years). The other general characteristics of the included studies are described in Appendix A in the Appendix A. Most of the included RCTs had a low risk of bias, except for 21 trials that showed a high risk of bias (Appendix A in the Appendix A). Qualitative syntheses of the included studies are presented in Appendix A in the Appendix A.

Compared with a placebo, ACE inhibitors were significantly associated with an increased risk of dry cough (RR = 2.66, 95% CI = 2.20 to 3.20, *p* < 0.001), hypotension (RR = 1.98, 95% CI = 1.66 to 2.35, *p* < 0.001), and hyperkalemia (RR = 1.24, 95% CI = 1.01 to 1.52, *p* = 0.037). However, we found no statistically significant association between ACE inhibitor exposure and the incidence of angioedema (RR = 1.32, 95% CI = 0.90 to 1.95, *p* = 0.160). Among the prespecified adverse outcomes of interest, the heterogeneity was low for all the outcomes, except for dry cough (*I*^2^ = 81.5%) (Table 1). Substantial heterogeneity for the dry cough outcome was resolved through subgroup analysis based on drug name (Appendix A in the Appendix A). Meta-regression did not suggest an ACE inhibitor dose dependent trend, or a treatment duration associated with the incidence of dry cough (B = 0.14, 95% CI = −0.11–0.39, *p* = 0.286; B = 0.00, 95% CI = −0.01–0.01, *p* = 0.631, respectively) (Appendix A in the Appendix A). The funnel plots followed by the linear regression test did not suggest the influence of small-study effects, including publication bias, for any of the outcomes except for hypotension (linear regression test, *p* = 0.005) (Figure 4). Sensitivity analyses did not suggest any significant changes in the summary effect estimates when studies with a high risk of bias were excluded (Appendix A in the Appendix A). The risk difference was quantified to be 0.037 (95% CI = 0.030–0.043, *p* < 0.001), 0.030 (95% CI = 0.021–0.039, *p* < 0.001), and 0.009 (95% CI = −0.001–0.019, *p* = 0.070) for dry cough, hypotension, and hyperkalemia, respectively.

An exploratory approach identified one more adverse outcome associated with ACE inhibitor exposure, i.e., dizziness (RR = 1.46, 95% CI = 1.26–1.70, *p* < 0.001). The heterogeneity was low (Table 1), and there was no evidence of small-study effects, including publication bias, for this outcome (linear regression test, *p* = 0.106). No significant changes in the summary effect estimates were found when sensitivity analyses were applied (Appendix A in the Appendix A). The risk difference was quantified to be 0.017 (95% CI = 0.009–0.025, *p* < 0.001). Overall, drug discontinuation probably due to adverse drug reactions did not statistically significantly differ between the ACE inhibitor group and the placebo group (4917/32,803 vs. 4361/30,661; Cochran’s *Q* test, *p* < 0.001, *I*^2^ = 60.2; RR = 1.09, 95% CI = 0.97–1.21, *p* = 0.143). Meta-regression did not find a dose-dependent relationship between ACE-inhibitor-dose levels and drug discontinuation (B = 0.02, 95% CI = −0.09–0.12, *p* = 0.758). Additionally, drug discontinuation was not significantly associated with treatment duration (B = 0.00, 95% CI = −0.00–0.01, *p* = 0.236).

## 4. Discussion

This meta-analysis of 257 RCTs allowed the precise estimation of the relative risk of ACE inhibitor-associated adverse outcomes. Analyzing all available safety data in multiple RCTs involving patients with a variety of diseases and conditions helped to characterize and quantify the prospect of ACE inhibitor-associated adverse outcomes. Overall, ACE inhibitor usage was significantly associated with an increased risk of dry cough, hypotension, dizziness, and hyperkalemia, with the number needed to harm of about 28, 33, 59, and 111, respectively. Although these ACE inhibitor-associated adverse outcomes have been well-recognized for decades, we provided overall safety profiles of ACE inhibitors across a wide range of clinical indications that can be useful to advise patients who are required to take ACE inhibitors.

As expected, dry cough is the most common adverse effect of ACE inhibitors and our meta-analysis found that ACE inhibitors significantly increased the risk of dry cough by about 2.6 times. Following ACE inhibitor treatment, the absolute excess risk of dry cough was 0.037 (95% CI = 0.030–0.043), that is, with approximately 28 patients (95% CI = 23–34) treated with ACE inhibitors, 1 patient would present with dry cough attributable to an ACE inhibitor. Based on meta-regression and subgroup analysis, ACE-inhibitor-related dry cough was not dose- or duration-dependent; perindopril and ramipril yielded the highest significant risk of dry cough (RR = 4.19 and RR = 4.13, respectively), while enalapril yielded the lowest risk of dry cough (RR = 1.23). The mechanism of ACE-inhibitor-induced dry cough is perceived to be related to a bradykinin-induced sensitization of airway sensory nerves and an accumulation of substance P in the respiratory tracts [30]. Although the dry cough may be considered merely a nuisance, it may necessitate additional hospital visits or drug discontinuation in more severe cases [31].

We observed increased risks of hypotension (RR = 1.98) and hyperkalemia (RR = 1.24) among individuals taking ACE inhibitors. The absolute risk increase was 0.030 and 0.009, respectively; that is, with approximately 33 and 111 patients who are treated with ACE inhibitors, 1 individual patient would present with hypotension and hyperkalemia, respectively, attributable to ACE inhibitor usage. These adverse drug reactions are attributable to the mechanism of action of ACE inhibitors related to reduced angiotensin II formation [32,33]. It was observed that hyperkalemia due to ACE inhibitor treatment was uncommon (with the number needed to harm of 111). Previous evidence suggested that such an adverse event often occurs when ACE inhibitors are concomitantly prescribed with potentially interacting drugs, such as spironolactone or chlorothiazide [34], or when patients have certain comorbidities, such as chronic kidney disease [35].

In our meta-analysis, ACE inhibitors were found to be statistically significantly associated with dizziness. Of every 59 patients (95% CI = 41–109) who are treated with ACE inhibitors, there will be approximately 1 individual complaining of dizziness attributable to ACE inhibitor treatment. Our finding is consistent with a reverse association study in which the authors reported ACE inhibitors to be one of the most common drug classes that were associated with dizziness among patients who presented with dizziness with an unknown cause [36]. Although the mechanisms underpinning ACE inhibitor-related dizziness remain uncertain, dizziness is likely a symptom of orthostatic hypotension [37].

In contrast with previous evidence [38,39], our meta-analysis does not suggest a significant increased risk of angioedema when individuals are exposed to ACE inhibitors (RR = 1.32, 95% CI = 0.90–1.95, *p* = 0.160). The discrepancy between our findings and previous evidence may be owing to angioedema occurring more commonly among certain populations (e.g., African Americans or heart failure patients) [40,41,42], while the RCTs included in our meta-analysis involved study populations with diverse ethnicities and underlying conditions. Furthermore, it is possible that the trials did not have a follow-up period long enough to detect the incidence of angioedema, because more than half of the cases showing this adverse consequence may occur after one year (or longer) of ACE inhibitor treatment [43,44]. As documented earlier, angioedema is rare but life-threatening [45,46,47], so healthcare professionals should still be aware of this possible adverse consequence when ACE inhibitors are prescribed to their patients [48].

There is a possibility that our findings based solely on published RCTs may misestimate the risk estimates of ACE-inhibitor-related adverse outcomes [49,50]. The inclusion of only published RCTs in a systematic review may suffer from reporting bias. Some articles provided only a brief description of adverse events or reported only statistically significant adverse event findings [51,52,53,54]. Nevertheless, the impact and significance of unpublished data beyond those reported in corresponding publications remain to be determined [55]. Furthermore, the inclusion of only RCTs in a systematic review and meta-analysis has certain limitations. For example, it is well-acknowledged that the RCT population usually differs from the real-world population [56,57]. Exclusion of certain groups of patients at imminent risk of serious harm in RCTs would potentially lead to a misestimation of actual drug risks. Lastly, we acknowledge that most of the included studies had relatively short durations of follow-up, so the adverse events presented in this study may not be representative of the long-term adverse consequences of ACE inhibitors [58,59,60].

## 5. Conclusions

In conclusion, in the present systematic review and meta-analysis, we characterized and quantifies the relative risk of several adverse outcomes due to ACE inhibitors across a diverse range of populations. Treatment with ACE inhibitors puts individuals at increased risk of dry cough, hypotension, dizziness, and hyperkalemia by 2.66, 1.98, 1.46, and 1.24 times, respectively. The overall findings of the present study help in guiding informed decisions about the management of diseases requiring ACE inhibitor therapy. These findings can also be used to advise patients requiring ACE inhibitor therapy. 

## Figures and Tables

**Figure 1 ijerph-19-08373-f001:**
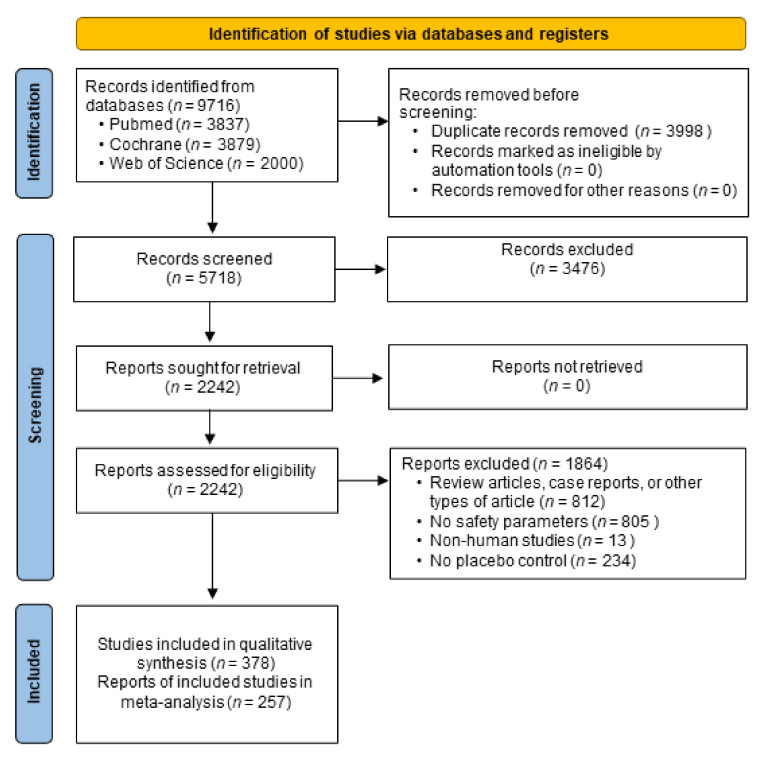
Flow diagram of this study.

**Figure 2 ijerph-19-08373-f002:**
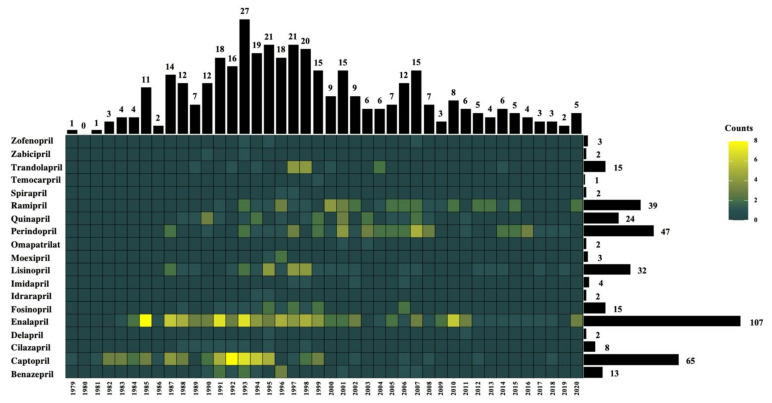
Heatmap analysis showing the number of randomized controlled trials of ACE inhibitors vs. placebo, published between 1979 and 2020.

**Figure 3 ijerph-19-08373-f003:**
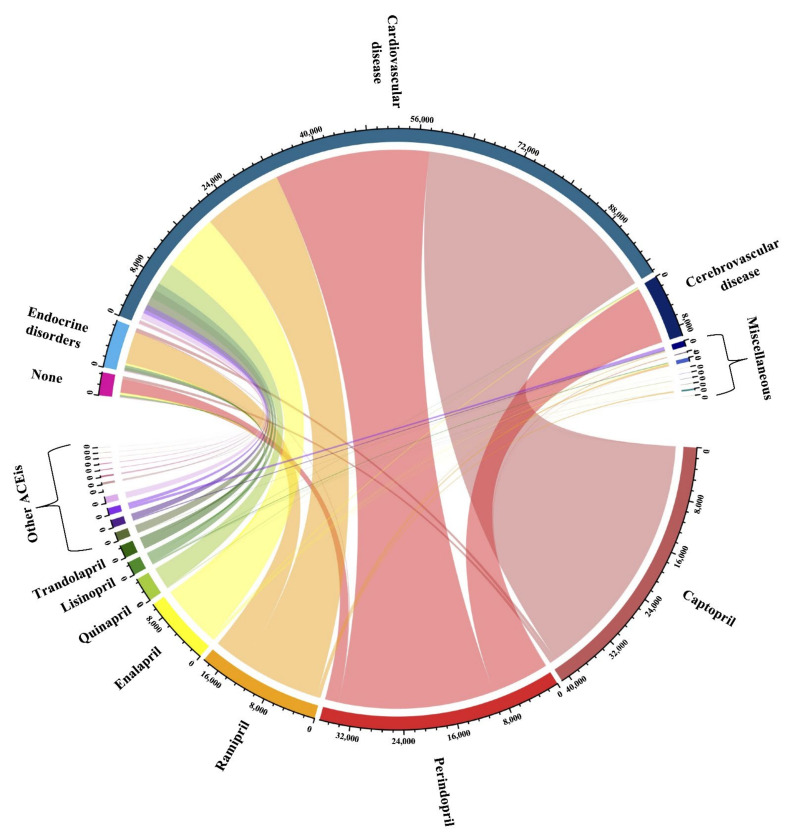
Chord diagram showing underlying conditions of research participants and ACE inhibitor administration.

**Figure 4 ijerph-19-08373-f004:**
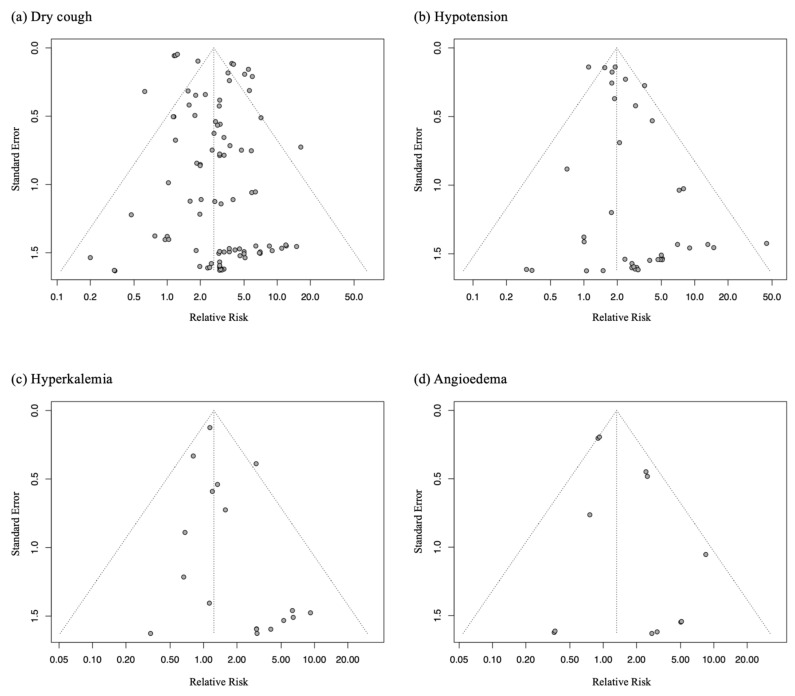
Funnel plots of the prespecified adverse outcomes of interest: (**a**) dry cough, (**b**) hypotension, (**c**) hyperkalemia, and (**d**) angioedema.

**Table 1 ijerph-19-08373-t001:** Meta-analysis of 257 randomized controlled trials of ACE inhibitors vs. placebo on adverse outcomes.

Adverse Outcome	Number of Studies	Intervention		Heterogeneity		Effect Estimates		
		ACE Inhibitor (Events/Total)	Placebo (Events/Total)	Chi	I^2^ (%)	RR	95% CI	*p*-Value
** *Confirmatory approach* **								
Cough	99	4352/54,518	2112/53,220	<0.001	81.5	2.66	2.20–3.20	<0.001
Hypotension	40	780/16,634	341/14,944	0.172	17.4	1.98	1.66–2.35	<0.001
Hyperkalemia	18	140/6331	96/6140	0.571	0.0	1.24	1.01–1.52	0.037
Angioedema	12	143/16,695	122/16,642	0.164	28.7	1.32	0.90–1.95	0.160
** *Exploratory approach* **								
** *Cardiovascular problems* **								
Cardiac arrhythmia	8	99/3562	96/3311	0.498	0.0	0.99	0.75–1.29	0.915
Chest pain	29	289/8522	294/8360	0.606	0.0	0.96	0.82–1.11	0.559
Congestive heart failure	11	184/3767	209/3788	0.793	0.0	0.88	0.74–1.05	0.169
Flushing	5	21/444	11/425	0.017	66.6	3.18	0.49–20.48	0.223
Hypertension	14	316/16,701	523/16,615	0.232	20.4	0.60	0.50–0.73	<0.001
Myocardial infarction	21	68/6398	57/5491	0.906	0.0	0.87	0.60–1.26	0.463
Palpitation or tachycardia	7	40/666	4/438	0.352	10.1	2.14	0.72–6.34	0.169
Reinfarction	4	15/472	15/473	0.271	23.4	0.87	0.27–2.81	0.820
Revascularization	5	258/4702	130/4006	0.061	55.7	0.82	0.55–1.25	0.358
Worsening heart failure	5	28/346	87/355	0.348	10.2	0.36	0.23–0.56	<0.001
** *Gastrointestinal problems* **								
Abdominal pain	10	37/507	7/334	0.480	0.0	1.65	0.74–3.67	0.220
Constipation	5	5/484	3/184	0.730	0.0	0.96	0.26–3.51	0.952
Diarrhea	9	70/3440	60/31	0.276	18.8	1.10	0.68–1.78	0.708
Flatulence	3	2/80	2/83	0.364	1.1	0.89	0.15–5.38	0.903
Gastroenteritis	4	4/301	6/182	0.316	15.3	0.61	0.16–2.33	0.467
Indigestion	3	5/233	2/148	0.106	55.4	1.29	0.09–17.68	0.847
Nausea & vomiting	22	102/2118	94/1437	0.744	0.0	0.82	0.63–1.07	0.139
Nonspecific gastrointestinal symptom	5	19/370	18/360	0.792	0.0	0.98	0.54–1.78	0.955
** *Integumentary problems* **								
Allergy	6	9/511	5/493	0.944	0.0	1.47	0.56–3.89	0.432
Cramp	4	3/369	2/158	0.298	18.5	0.91	0.17–4.87	0.910
Joint & muscle pain	6	33/457	10/255	0.254	24.0	1.87	0.79–4.45	0.154
Skin infection	4	23/267	26/256	0.248	27.3	1.04	0.32–3.37	0.947
Skin rash	23	52/2264	34/1734	0.970	0.0	1.20	0.79–1.83	0.394
** *Neuropsychiatry problems* **								
Anxiety	3	6/242	10/250	0.285	20.4	0.80	0.20–3.18	0.753
Dizziness	59	566/37,879	267/35,753	0.481	0.0	1.46	1.26–1.70	<0.001
Headache	66	857/383,551	726/35,699	0.293	8.1	0.95	0.82–1.09	0.460
Sleep disturbance	5	19/134	8/132	0.432	0.0	1.77	0.80–3.91	0.159
Stroke	6	7/1391	9/1282	0.772	0.0	0.73	0.27–1.93	0.522
Syncope	10	40/1630	21/1371	0.608	0.0	1.49	0.89–2.49	0.134
** *Respiratory problems* **								
Dyspnea	4	2/342	17/285	0.358	7.0	0.28	0.06–1.25	0.095
Respiratory infection	9	100/1366	58/949	0.306	15.3	1.27	0.89–1.81	0.194
Respiratory problems	10	46/1158	20/671	0.563	0.0	1.33	0.77–2.29	0.303
** *Renal problems & electrolyte imbalance* **								
Edema	11	67/4104	77/3978	0.348	10.1	0.81	0.54–1.19	0.281
Hypokalemia	5	7/309	24/307	0.106	47.5	0.47	0.11–2.00	0.307
Proteinuria	7	75/11,824	88/11,818	0.302	16.8	0.76	0.48–1.20	0.243
Renal impairment	19	432/43,013	314/42,851	<0.001	63.3	1.13	0.72–1.75	0.598
Urination	4	5/482	3/270	0.162	41.61	0.92	0.14–6.11	0.933
** *Miscellaneous* **								
Cancer	11	76/4833	68/4707	0.468	0.0	1.09	0.79–1.52	0.588
Erectile dysfunction or impotence	4	3/776	5/322	0.485	0.0	0.36	0.09–1.45	0.151
Fatigue	36	194/3542	105/2307	0.430	2.3	1.18	0.94–1.49	0.162
Flu symptoms	3	11/119	3/93	0.165	44.5	0.81	0.08–8.16	0.861
Inflammation	5	24/250	13/219	0.816	0.0	1.68	0.94–3.01	0.082
Malaise	4	9/149	4/118	0.780	0.0	1.64	0.54–4.92	0.381
Neutropenia	4	17/761	21/754	0.834	0.0	0.80	0.44–1.46	0.464
Stress	3	8/127	5/116	0.539	0.0	1.31	0.43–3.97	0.629
Sweating	3	1/223	4/110	0.428	0.0	0.31	0.06–1.68	0.172

## Data Availability

All data used to support the findings of this study are available from the corresponding author upon reasonable request.

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
