# Peer review of "Adverse Effects of Angiotensin-Converting Enzyme Inhibitors in Humans: A Systematic Review and Meta-Analysis of 378 Randomized Controlled Trials"

_ijerph, 2022, doi:10.3390/ijerph19148373_

Round 1

Reviewer 1 Report

Takuathung et al reported a correlative analysis of ACE inhibitors to various adverse reactions from various RCTs. The article is very well written and some of the figures are outstanding. I do have a few comments as follows:

1. The RCTs that were included in the study, and the patients that were included in those studies, were they only on ACE inhibitors and no other medications? How the authors can undoubtedly claim that the adverse reactions such as dry cough, hyperkalemia, and angioedema are only and only caused as an adverse reaction of ACE inhibitors and not due to any other medications, or disease or lifestyle practices? For example, on page 11 line 229-231 the authors said that hyperkalemia is an uncommon adverse reaction of ACE inhibitors and polypharmacy i.e. ACE inhibitors and spironolactone or chlorothiazide and patient with CKD have an increased possibility of having hyperkalemia. I am worried about how confidently these adverse reactions can be claimed as a result of only and only ACE inhibitors’ intake

2. Is it possible to provide a better description of the funnel plot on figure 4.

Author Response

Thank you for your positive comment. This study was designed to include only randomized-controlled trials comparing an ACE inhibitor and a placebo. A randomized, placebo-controlled design is considered the ‘gold’ standard for testing intervention and provides the strongest possible evidence of causation. Therefore, our analysis can suggest ACE inhibitor-associated adverse outcomes. Also, we have revised the description of the funnel plot in Figure 4 as follows: “Figure 4 Funnel plots of the pre-specified adverse outcomes of interest: (a) dry cough, (b) hypotension, (c) hyperkalemia, and (d) angioedema.” Thank you for your suggestion and careful consideration.

Reviewer 2 Report

The manuscript under the title "Adverse Effects of Angiotensin-Converting Enzyme Inhibitors 2 in Humans: A Systematic Review and Meta-Analysis of 378 3 Randomized Controlled Trials",  can be accepted after revising the following points"

1- There are many abbreviations in the manuscript so adding an abbreviation section would ease its reading.

2- well written and valuable findings can be accepted after minor changes.  

Author Response

We have added an abbreviation section in the revised manuscript. It now reads “Abbreviations: ACE, angiotensin-converting enzyme; CI, confidence interval; PRISMA, Preferred Reporting Items for Systematic Reviews and Meta-Analyses; RCT, randomized controlled trial; RoB2, revised Cochrane risk-of-bias tool for randomized trials; RR, relative risk.” Thank you very much for your positive comment and careful consideration.

Reviewer 3 Report

The manuscript is both well-written and interesting. I have no hesitation in recommending publication since a large dataset has been analysed, apparently with statistical rigour and the results should therefore be disseminated. 

The statistics involved are largely outside my area of expertise, as a structural biologist, but it seems that the widely reported hyperkalemia and angioedema effects are really not of massive signifcance. Also, and again my ignorance of the methods involved, I was slightly confused as to why the reports of hypertension and worsening heart failure have about the same significance as dry cough and hypotension, despite the number of cases being very much lower. Perhaps some additional explanation of these effects would help the non-specialist reader.

Author Response

Thank you for your positive and valuable comment. ACE inhibitors are the first-line drug for the treatment of hypertension and heart failure with reduced ejection fraction. Therefore, it is not surprising to see their effects on a decreased risk of hypertension (RR = 0.60, p <0.001) and worsening heart failure (RR = 0.36, p <0.001). Thank you for your careful consideration.

Reviewer 4 Report

The authors present a systematic review and meta-analysis regarding adverse effects of ACE inhibitors. The work is very well-organized, it is  related with the SI's aim and scope and it is very-well presented. Although the ADEs that are found to be related with the ACE inhibitors are known the present works provides some additional overall insights summarizing existing RCTs. Table 1 could have been better presented in my opinion. Apart of that the work can be further processed. 

Author Response

Thank you for your positive comment. In the revised manuscript, we have edited the format of Table 1 to be easier to read. Thank you for your careful consideration.